# The First 1000 Days of Life: How Changes in the Microbiota Can Influence Food Allergy Onset in Children

**DOI:** 10.3390/nu15184014

**Published:** 2023-09-16

**Authors:** Veronica Notarbartolo, Maurizio Carta, Salvatore Accomando, Mario Giuffrè

**Affiliations:** 1Neonatal Intensive Care Unit with Neonatology, “G.F. Ingrassia” Hospital Unit, ASP 6, 90131 Palermo, Italy; veronicanotarbartolo@gmail.com; 2Neonatology and Neonatal Intensive Care Unit, University Hospital Policlinic “Paolo Giaccone”, 90127 Palermo, Italy; maurizio.carta@policlinico.pa.it; 3Department of Health Promotion, Mother and Child Care, Internal Medicine and Medical Specialties, University of Palermo, 90127 Palermo, Italy; salvatore.accomando@unipa.it

**Keywords:** gut microbiota, human milk oligosaccharides (HMOs), complementary feeding, food allergy (FA), breastfeeding, first 1000 days of life, dysbiosis, newborns

## Abstract

Background: Allergic disease, including food allergies (FA)s, has been identified as a major global disease. The first 1000 days of life can be a “window of opportunity” or a “window of susceptibility”, during which several factors can predispose children to FA development. Changes in the composition of the gut microbiota from pregnancy to infancy may play a pivotal role in this regard: some bacterial genera, such as *Lactobacillus* and *Bifidobacterium*, seem to be protective against FA development. On the contrary, *Clostridium* and *Staphylococcus* appear to be unprotective. Methods: We conducted research on the most recent literature (2013–2023) using the PubMed and Scopus databases. We included original papers, clinical trials, meta-analyses, and reviews in English. Case reports, series, and letters were excluded. Results: During pregnancy, the maternal diet can play a fundamental role in influencing the gut microbiota composition of newborns. After birth, human milk can promote the development of protective microbial species via human milk oligosaccharides (HMOs), which play a prebiotic role. Moreover, complementary feeding can modify the gut microbiota’s composition. Conclusions: The first two years of life are a critical period, during which several factors can increase the risk of FA development in genetically predisposed children.

## 1. Introduction

Allergic disease has been identified by the World Health Organization (WHO) as a major disease on a global scale. Allergic diseases include asthma, food allergies (FAs), allergic rhinitis, and eczema. These are traditionally referred to as type 1 hypersensitivity reactions [1]. The term “sensitization” refers to a process in which T and B cells respond to an allergen, leading to the production of specific IgE antibodies. After a new allergen exposure, sensitization may or may not be associated with a clinical allergic reaction [2]. Over the last two decades, allergic diseases’ prevalence in childhood has been increasing [3]. In the USA, 1 in 13 children suffers from FA [4]. FA results from an abnormal response of the immune system to food antigens and can be life-threatening in children or impact their quality of life [4]. FA can be either IgE-mediated or not, and it can also be both; nowadays, IgE-mediated FA is the best characterized and it is the classically recognized type in society [5]. Allergens are able to stimulate the release of local mediators, such as histamines, by cross-linking IgE–IgE receptor complexes on the surfaces of mast cells and basophils [6]. Allergic diseases are associated with an increase in the T-helper_2_ (Th_2_) cell response. During pregnancy, the fetal Th_1_ immune response is suppressed to prevent excessive responses to maternal antigens; at birth, the Th_2_ response to novel antigens increases. Exposure to the gut microbiota shifts this response to the development of Th_1_ cells, with a consequent promotion of immune tolerance and T-cell maturation [2,7]. The gut microbiota plays a key role in modulating and influencing immune tolerance [8], and early microbial exposure is highly important for children’s immune development [4]. According to the “hygiene hypothesis”, proposed by Dr. David Strachan in the 1980s [9], exposure to infection in the first months of life can be protective against allergic diseases. An early-life microbial experience is associated with an improvement in immune development and with a reduced risk of the onset of allergic diseases [2,10]. In fact, a farm environment with a diverse range of environmental microbes to which children can be subjected is associated with a reduced risk of allergic diseases, especially asthma [11]; on the contrary, an excessively sanitized Western lifestyle can disrupt children’s gut microbiota and disturb normal immune system development [12]. The “hygiene hypothesis” was later expanded by the “microflora hypothesis” in 2005, which indicates that an early disturbance of infants’ gut microbiota could alter immunological tolerance and lead the immune system to tend toward a state of allergic disease [1]. A dysbiosis of the gut microbiota is, in fact, associated with increased intestinal permeability, an aberrant inflammatory response, and a different composition of the microbiota [8]. Generally, it has been shown that intestinal dysbiosis is correlated with a reduction in *Lactobacilli* and *Bifidobacteria*, which seem to protect against the development of atopy by promoting the Th_1_ immune response and inhibiting the Th_2_-type immune response. On the contrary, early colonization by *Clostridium difficile* and *Staphylococcus aureus* is more likely to occur in children who develop an allergy later in life [8]. In this context, breastfeeding plays a pivotal role in modulating the immune system development of newborns and in the establishment of a healthy gut microbiota [13]. Several factors can influence and alter the composition of pregnant people’s and newborns’ gut microbiota: the mode of delivery, feeding practices, hygienic and social status, gestational age, the use of antibiotics, and maternal and infant diseases [14]. The maternal gut and breast and the infant gut are linked by the so-called “gut–breast axis”, a gastrointestinal tract that, when it works correctly, is fundamental to the assembly of human milk components and the modulation of the development of the newborn’s gastrointestinal tract, with lifelong effects [15]. The aim of this narrative review is to summarize the main changes in the microbiota in the first years of life, during the fetal, neonatal, and early childhood periods, and to describe how these modifications may negatively predispose children to the onset of FA.

## 2. Methods

A narrative review was performed according to the most recent available literature (2013–2023). Original papers, clinical trials (both in humans and animals), meta-analyses, and reviews in English were included. Case reports, series, and letters were excluded. The research was conducted using the following keywords (alone or in combination): children, childhood, infants, newborns, gut microbiota, human milk, breastfeeding, complementary feeding, food allergy, diet, pregnancy, epigenetics, allergy, oligosaccharides, dysbiosis. PubMed and Scopus were used as the electronic databases.

## 3. Epigenetics and the Gut Microbiota

“Microbiota” is a term that refers to all of the microorganisms present within a niche in the human body, which have a symbiotic relationship with the host; it includes bacteria, viruses, fungi, parasites, and *archaea*. On the other hand, “microbiome” refers to the genetic material from the “microbiota” [16,17]. The bacterial component is the most prevalent (from 10^13^ to 10^14^ organisms) [2]. Bacterial communities are found in the gastrointestinal and urogenital tract, skin, and oral cavity, but the gut microbiota is the most studied [18]. The two major characteristics of the gut microbiota are richness and diversity; the first term refers to the total number of bacterial species present, the second refers to the number of individual bacteria from each of the bacterial species present [19]. It is possible to differentiate between groups at two levels: the alpha and beta diversity [20]. In the context of the human microbiota, alpha diversity is related to the compositional complexity of a single sample, whereas beta diversity is associated with the taxonomical differences between samples [21]. Atopy also seems to contribute to gut diversity [8]. Continuous exposure to allergy-protective microorganisms, especially during the prenatal and postnatal periods, results in a form of innate and adaptive immune training that makes individuals resistant to danger signals from the environment. This is the so-called “hypothesis of inflammatory resilience”. A lack of biodiversity is, in fact, associated with an inappropriate and exaggerated inflammatory response to danger signals [18]. The allergic process is the result of gene–environment interactions with epigenetic changes in one or more susceptibility genes [18]. Diet is able to influence the composition of the microbiota via different mechanisms: by regulating intestinal barrier function, modifying the composition of the bacterial gut microbiota, and influencing gene expression, intestinal motility, and the immune system response [22,23]. Additionally, microbiota-derived metabolites can be used as epigenetic substrates [24]. Epigenetic mechanisms are able to modify gene expression without altering deoxyribonucleic acid (DNA) sequences. The major epigenetic processes involved are DNA methylation patterns, histone modification, chromatin remodeling, and micro-ribonucleic acid (RNA) (miRNAs), even if there is still some debate as to whether miRNAs can be considered a real epigenetic phenomenon [19,25,26,27]. *Bifidobacteri* and *Lactobacilli* are able to produce folate, an essential molecule involved in methylation processes; changes in bacterial composition can shift the DNA methylation status of the host. At the same time, short-chain fatty acids (SCFAs), produced by commensal microbes during fermentation patterns, are able to influence histone modifications [24]. A pea-protein-rich diet is associated with an increased level of SCFAs, which are highly important in maintaining the integrity of the mucosal barrier [22]. Contextually, in murine models, a saturated-lipid-rich diet has been demonstrated to induce the expression of pro-inflammatory genes’ patterns (i.e., Toll-like receptors, TLRs), associated with intestinal dysbiosis [28]. A systematic review conducted by Hartwig et al. [29] has shown that breastfeeding might influence DNA methylation, although more studies are needed. For example, it has been demonstrated that the duration of breastfeeding may be associated with DNA methylation in children. In particular, the hypermethylation of the gene encoding sorting nexin 25 (SNX25) is related to an increase in transforming growth factor (TGF)-β activity, a cytokine that promotes immunoglobulin A (IgAs) secretion [6]. Moreover, a recent review conducted by Esch et al. [30] demonstrated that allergic diseases are associated with changes in DNA methylation patterns in Th_1_, Th_2_, Th_9_, Th_17_, and T regulatory (Treg) cells. In particular, there would be an alteration in forkhead box P3 (FoxP3+) demethylation, the molecule involved in tolerance induction [30,31]. At the same time, milk-derived miRNAs are able to regulate gene expression in a dose-dependent manner [32]. MiRNAs are short noncoding RNA molecules that are able to induce mRNA degradation and/or the post-transcriptional inhibition of translation. Some of them are involved in the maturation of Treg cells [30]. Nevertheless, data regarding the role of breastfeeding’s epigenetic mechanisms and its preventative role on the development of allergies are very limited [6], and further data are needed. It is quite clear that the digestion and metabolism of molecules introduced with the diet can influence human health depending on the individual genotype (“nutrigenetics”) and gene expression (“nutritional epigenetics”) [27]. The regulation of this last one via epigenetic mechanisms (i.e., environmental and dietary exposure) plays a particular role in FA [4,26].

## 4. First 1000 Days of Life: “A Window of Opportunity” in the Gut Microbiota

The first months of life are fundamental in the establishment of a healthy gut microbiome, whereby several factors can influence its normal composition and function. In this period, unhealthy cues can cause alterations in gene expression later in life, increasing the risk of multifactorial environmentally driven diseases such as allergies [19]. In fact, according to Barker’s theory [33], the first 1000 days after conception are a critical period in which most human development takes place [27]. During this time frame, which comprises pregnancy, the neonatal period, and the first two years of life, some factors may have a beneficial effect (“window of opportunity”), while others may increase the possibility of diseases developing (“window of susceptibility”). See Figure 1.

In this timeframe, epigenetic intervention (the “window of intervention”) might prevent or even reverse the negative effects of environmental risk factors [6]. Generally, vaginal delivery, breastfeeding, a rural environment, exposure to pets, and a fiber-enriched diet seem to be protective factors against the onset of FA. To date, no specific bacterial taxa have been associated with the onset of FA [34]. In any case, normal microbiota development involves early colonization by facultative anaerobes (especially those belonging to the *Enterobacteriaceae* family), which reduce initial oxygen supplies to create a favorable environment for anaerobe colonization (especially those belonging to the *Bacteroidaceae* family). Members of *Bacteroidaceae* families are able to stimulate the production of mucin, which maintains an intact gut microbiota–mucin barrier. The *Enterobacteriaceae*/*Bacteroidaceae* (E/B) ratio tends to decline over time: its persistent elevation is very frequent in food-sensitized infants, as a marker of the delayed maturation of the gut microbiota [35] and as a marker of a less mature microbiome [4]. SCFAs, such as acetate, butyrate, and propionate, derive from bacteria fermentation and seem to have many beneficial effects in autoimmune and inflammatory diseases [34]. Among these, butyrate is the most important: it might contribute to the development of immune oral tolerance and to the prevention and treatment of FA by modulating Treg activity [26]. Butyrate inhibits the release of pro-inflammatory cytokines and promotes anti-inflammatory responses [19]. Demonstrating this phenomenon, levels of butyrate tend to be lower in children affected by a cow’s milk allergy (CMA) at one year [4,34]. Moreover, Roduit et al. [36], have shown that a high butyrate concentration in children’s stools at one year of age is associated with a significant reduction in the atopic sensitization to food and/or inhalant allergens. At the same time, children were less likely to have a diagnosis of FA or allergic rhinitis between three and six years of age. Therefore, changes in the composition of the gut microbiota during the first months of life could impair the future health of children.

### 4.1. Gut Microbiota of Pregnant Women

The environment already plays a crucial role in modulating the composition of the gut microbiota of newborns during the fetal period [3]. It is likely that nutritional exposure during the immune programming period may play a role in FA development [37]. The role of the maternal diet has been thoroughly evaluated: a prospective cohort study conducted by Brzozowksa et al. [3] showed that a reduced intake of vitamin C and magnesium during pregnancy can be associated with a greater risk of developing wheezing in the first two years of life. Moreover, an increased intake of vitamin D, retinoic acid, zinc, and copper may positively influence epigenetic patterns associated with allergic diseases in children [3,6,10,26,37]. At the same time, the perinatal supplementation of polyphenols seems to prevent allergies in offspring [6]. The immunomodulatory role of vitamin D is well established: lower plasmatic levels are associated with an increased risk of the development of allergic diseases [5]. Nevertheless, previous studies have shown a greater risk of overall allergies in children born by vitamin-D-supplemented mothers [38,39]. It is likely that there is a U-shaped relationship between vitamin D levels and the risk of allergies: both too little and too much vitamin D correlate with the greatest risk [5,40]. In animal studies, vitamin A supplementation during pregnancy was a possible intervention for allergy prevention in the neonatal stage of life [41]. Additionally, *n*-3 long-chain polyunsaturated fatty acids (LC-PUFAs) may modulate the development of IgE-mediated allergic disease and regulate immune responses by influencing the Th_1_/Th_2_ balance in infants [26,37]. In particular, in an observational study conducted by Best et al. [42], it was hypothesized that increased *n*-3 LC-PUFA intake during pregnancy could be associated with a reduction in the prevalence of childhood allergic diseases [43]. In fact, ω-3 PUFAs are able to inhibit Th cell differentiation, reducing the risk of the development of allergies [10]. An increase in PUFA levels is correlated with an abundance of *Holdemania* spp. in maternal feces during pregnancy: it is a Gram-positive anaerobic bacterial genus associated with a reduced risk of FA in offspring and it can be used as a predictor marker [44]. Nevertheless, if it is true that *n*-3 LC-PUFA supplementation in pregnancy is associated with a reduced risk of allergy in children, it has not shown a positive influence when children are directly supplemented. This may be because pregnancy is an important time that influences the development of the immune system more than the early postnatal period [30]. Since it has been shown that breastfeeding plays a protective role against the onset of FA due to the significant levels of butyrate contained therein, it might be useful to increase human milk butyrate concentrations via the modulation of maternal diet [26,34]. High levels of vegetable consumption by pregnant people seems to be associated with a reduced risk of allergic disease in offspring, due to the increased diversity and richness of the gut microbiota (*Holdemania*, *Roseburia*, *Lachnospira*, and *Coprococcus* spp.) [11]. In a prospective population-based cohort study conducted by Tuokkola et al. [45], it was demonstrated that diet during pregnancy may play a more significant role in influencing the development of FA than the lactation period, suggesting antigen-specific induction of tolerance in subjects who are not genetically predisposed to the development of allergic disease. Nevertheless, no clear results have been established, so no recommendations for clinical practice are available [3,10,41,45]. In fact, according to the most recent studies, the American Academy of Pediatrics (AAP) and the European Food Safety Authority (EFSA) [46] have concluded that there is not sufficient evidence to support maternal dietary restriction during pregnancy or lactation to prevent atopic diseases in offspring [4]. Moreover, antibiotic exposure during pregnancy might play an important role in influencing the gut microbiota composition of the offspring and, consequently, the prevalence of FA, eczema, and asthma [11]; in murine models, it has already been shown that the use of antibiotics in pregnancy might enhance food sensitization in offspring [35]. For example, *Prevotella copri* is the predominant *Prevotella* species within the human gut microbiota: maternal antibiotic exposure is associated with its reduction and, consequently, with slight protection against allergic diseases [11]. In fact, during pregnancy, a maternal gut microbiota enriched by *Prevotella* spp. has a protective effect on FA, independently from the concentration of *Prevotella* in the offspring’s gut microbiota [9,44]. Antibiotic use during pregnancy has been associated with a reduction in the *Bifidobacterial* count in the neonatal gut during the first month of life [2]. Additionally, the environment where the pregnant person lives is important: children whose mothers live in a farm environment have an increased number of Treg cells due to a higher level of stimulatory bacterial lipopolysaccharide (LPS) variants [47]. In a recent systematic review conducted by Venter et al. [48], an index of maternal diets during pregnancy was proposed, which can be used to predict the risk of development of allergic diseases in offspring. The frequencies with which a person consumes vegetables, yogurt, fried potatoes, rice, red meats, fruit juice, and cold cereals were used to obtain data that were used to derive the predictive index for the development of allergic pathologies in offspring. Vegetables and yogurt seem to be protective factors against allergy development; on the other hand, fried potatoes, rice, red meats, fruit juice, and cold cereals can be risk factors [48]. These studies demonstrate, once again, that diet can influence the microbiota’s composition: for example, Asnicar et al. [49] have shown that a higher intake of vegetables and yogurt is associated with a more diverse microbiome and with higher levels of fecal butyrate, with protection from the development of allergies up to school age. At the same time, Russell et al. [50] found that a high protein/low carbohydrate diet correlated with a reduction in *Roseburia* spp. and with decreased fecal butyrate levels. More interestingly, a high intake of natural sugars (i.e., glucose, fructose, and sucrose) is associated with increased levels of *Bifidobacteria* and reduced levels of *Bacteroides*; conversely, artificial sweeteners seem to induce the opposite results [51,52]. Nevertheless, further studies are needed.

### 4.2. Gut Microbiota in the First Months of Life

The colonization of the neonatal gut by the microbiota constitutes a highly vulnerable period [14,53]. The translocation of maternal microorganisms (i.e., through placental tissue, the vagina, the maternal gut, and the *meconium*) is the starting point for the establishment of infants’ gut microbiotas [54]. At first, there is a prevalence of strict anaerobic bacteria such as *Bifidobacteri*, *Clostridi*, and *Bacteroides* [14]. Although the existing studies are contradictory, sensitization to milk and egg allergens is twice as likely to occur in children born by cesarean section (CS) [11,35]. In fact, the type of delivery correlates with a different composition of the gut microbiota: newborns born via vaginal birth display a higher abundance of *Bifidobacteri*, *Bacteroides*, *Lactobacilli*, and bacteria belonging to the *Lachnospiraceae* family; meanwhile, bacteria belonging to the *Enterococcaceae* and *Enterobacteriaceae* families are most abundant in CS-delivered newborns [54]. Vaginal delivery is associated with a healthier gut microbiota composition when compared to CS delivery. Moreover, maternal vaginal bacteria (i.e., *Lactobacillus* and *Prevotella* spp.) are prevalent in infants born vaginally, whereas microbes of maternal skin (i.e., *Staphylococcus* and *Propionibacterium* spp.) are most common in infants born by CS [19]. It is interesting that both the mode and the place of delivery can play a pivotal role: it has been demonstrated that, in children with a positive family history for atopy, a vaginal home delivery is more protective than a vaginal hospital one [8]. Atopic dermatitis is associated with lower gut microbiota diversity and the relative abundance of *Bacteroides* spp. by one month; an early colonization by *Clostridium difficile* at one month could be predictive of atopic sensitization at two years [35,55,56]. At the same time, anaerobes’ abundance is most frequent in infants with a confirmed allergy to cow’s milk. At three months, lower microbiota richness is a risk factor for food sensitization at one year of age; on the other hand, it has been shown that richness at one year is not associated with food sensitization. In fact, the critical period for microbiota development is “early infancy”. Food-sensitized infants exhibit an elevation in bacteria belonging to the *Enterobacteriaceae* family (i.e., *Escherichia, Shighella* spp.), even if there is a concomitant general reduction in *Proteobacteria*, that is, the phylum containing *Enterobacteriaceae* [35]. This apparent contradiction may be due to the concomitant reduction in other bacterial families belonging to the *Proteobacteria* phylum [55,57]. At the same time, in this slice of the population, a lower relative abundance of the *Bacteroidaceae* bacterial family has been shown [26,35]. Nevertheless, in a recent population-based cohort study, the composition of the gut microbiota of the first-pass *meconium* seems not to be related to later atopic manifestations in children [58]. The increasing use of antibiotics in childhood is a topic of concern: in a case–control study conducted by Hirsch et al. [12], it was shown that there is an association between antibiotic orders and several independent allergy diagnoses, especially regarding macrolides, in a dose–response relation. This is probably due to the capacity of antibiotics to alter the composition of the gut microbiota, with consequential functional changes that promote the development of allergic diseases. This effect seems to persist for a long period: penicillin and cephalosporins are more strongly associated with FA in the first two years of life, while macrolides are associated with FA later in childhood [59]. The increasingly prevalent use of antibiotics from the first months of life can affect the composition of the gut microbiota and cause “dysbiosis”, predisposing children to develop FAs (see Section 5).

#### 4.2.1. Role of Human Milk Microbiota

For a long time, human milk was thought to be sterile; recently, however, many studies have demonstrated that it constitutes a rich source of microbes, changes in which can influence children’s health. Two main origins of milk’s microbiota are known: the retrograde flow, i.e., the epiphenomenon of microbes’ transmission from the oral cavity of infants into the mammary duct during suckling, and the entero-mammary pathway, that is, the consequential translocation of maternal gut microbiota through the intestinal epithelial barrier [60]. The nine genera that constitute the “core” bacteriome of the human milk microbiota are *Staphylococcus*, *Streptococcus*, *Serratia, Pseudomonas*, *Corynebacterium*, *Ralstonia*, *Propionibacterium*, *Sphingomonas*, and *Bradyrhizobium*. They represent approximately half of the microbial milk community, although their abundance may vary between milk samples [17,61,62]. The “Mother–Human Milk–Infant” triad underscores the close connection between the pregnant person and the newborn, via human milk, in modulating the trajectory of infant development [63]. Breast milk provides a quarter of the intestinal microbiota of infants and its composition could influence the development of FA in children. It has been shown that a relative abundance of *Prevotella* spp. in the breast milk microbiota is more common in mothers whose children will not develop FA [64]. Moreover, it has been shown that, in the breastmilk of infants with allergic symptoms, there is a relative abundance of *Proteobacteria*, especially *Acinetobacter* and *Pseudomonas* spp. [64]. Nevertheless, data regarding the protective role of human milk against FA development remain contradictory [65].

#### 4.2.2. Role of Human Milk Oligosaccharides (HMOs)

Human milk is the golden standard for the nutrition of newborns that can impact FA development. The “entero–mammary link” is an active connection between the immune tissues in the maternal gut and the mammary glands; it results in a human-milk-specific IgA profile that can influence infants’ microbiome composition [41]. In early postnatal life, IgA deficiency might be associated with a higher risk of atopic dermatitis [2]. Human milk is a dynamic bio-fluid that contains a wide range of macro- and micro-nutrients. In this regard, human milk oligosaccharides (HMOs) have gained considerable attention, especially for their prebiotic role in stimulating the growth of *Bifidobacteria*, the dominant bacterial genus in breastfed infants, along with *Lactobacilli* and bacteria belonging to the *Enterobacteriaceae* family [41]. Generally, in breastfed infants, there is an abundance of *Bacteroidota*; meanwhile, *Firmicutes* are prevalent in formula-fed infants [54]. A reduced number of *Bifidobacterium* spp. is associated with the development of atopic diseases later in life [14,66]. In fact, *Bifidobacteria* are able to induce mast cell apoptosis, reducing allergic symptoms [54]. In particular, *Bifidobacterium infantis* is able to digest HMOs (“inside eater”) [67], producing metabolites that support infant development at the cellular level [66]. *Bifidobacterium infantis* is able to produce indole-3-lactic acid, a metabolite that decreases enteric inflammation and exerts regulatory effects on Th_2_ and Th_17_ cells [4]. In a recent study conducted by Dai et al. [66], it was pointed out that *Bifidobacterium infantis* is able to maintain the diversity of the gut microbiota after antibiotic exposure, making children less vulnerable to the onset of asthma. In fact, microbial diversity seems to be inversely associated with the risk of developing allergic pathologies [68]. Nevertheless, it has been shown that a greater microbial diversity is more beneficial in adults than in children [4]. Breastfeeding, which is fundamental for microbial development, is generally correlated with a less diverse microbiome [4]. HMOs are also able to play a direct role in promoting immune system maturation [14]. Approximately 200 different HMOs have been characterized: they present a lactose core that can be elongated by glucose or galactose or *N*-acetylglucosamine or fucose or sialic acid monosaccharides. They are not digested in the upper parts of infants’ gut tracts, so HMOs can reach the colon and be used as a substrate by microbes. Nowadays, cow-milk-derived infant formulas are often supplemented with non-digestible carbohydrates (galacto-oligosaccharides, GOS; fructo-oligosaccharides, FOS) to substitute HMO functions [14]. Moreover, a specific HMO profile in human milk has been shown to be associated with a reduced risk of food sensitization in the infant (for example, lacto-*N*-fucopentaose III, LNFP-III, would play a protective role) [41]. Another molecule contained in human milk that may play a potential role in preventing FA development is TGF-β, a cytokine with potent tolerogenic properties, although the relevant studies are not conclusive [41,69]. Further studies are necessary to evaluate the real link between FA and breastfeeding.

### 4.3. Gut Microbiota during Complementary Feeding

Gut bacterial colonization during infancy is a crucial event that establishes oral tolerance and establishes a functional digestive tract [70]. Before weaning, the gut microbiota is generally enriched by lactate-producing bacteria; meanwhile, when solid foods are introduced in the diet, there is an increase in the number of bacteria that are able to use a larger variety of carbohydrates and vitamins [70]. During complementary feeding, the gut microbiota is characterized by higher diversity in terms of bacteria species, especially *Enterococci*, *Enterobacteria*, *Clostridi*, and *Bacteroides* [26,41], and by a relative abundance of anaerobic bacteria [2]. The fluctuations in the abundance of *Actinobacteria* and *Firmicutes* in the first six months of life seem to be beneficial for the prevention of FA [44]. Moreover, progression in complementary feeding is associated with an abundance of *Lachnospiraceae* and *Ruminococcaceae* spp., and with a reduction in *Bifidobacterium* spp. This is the result of a transition from a breast-milk-promoted *Bifidobacteria* gut community toward a fiber- and protein-promoted gut microbial community (which is more diverse) [71]. The timing of the introduction of complementary feeding is important in terms of influencing the diversity of the infant gut microbiota [72]. An early introduction of solid foods (before three months vs. later) is associated with higher gut microbiome diversity and an increase in fecal SCFA composition (especially butyrate) at 12 months, as demonstrated by Differding et al. [73].

According to recent research, higher fecal concentrations of butyrate and propionate in adults may be related to worse metabolic outcomes; meanwhile, higher serum levels of SCFAs may be related to better health [74,75]. Regardless, the existing studies are not conclusive about infants. A prospective longitudinal study conducted by Pannaraj et al. [76] demonstrated that early weaning is associated with a faster maturation of the gut microbiome composition: for example, there would be a reduction in *Bifidobacteria* spp. and an increase in more adult-associated bacteria (i.e., genus *Bacteroides*) [77]. The establishment and development of the infant microbiome continues even after complementary feeding, up to three years of life, when the child’s gut microbiota appears more similar to the adult’s one [2]. Recent works suggest that a typical adult pattern of the gut microbiota may not be established before adolescence [65]. The delay in introducing allergenic foods during complementary feeding (beyond four to six months of life) is not recommended [4,69]. In this context, the AAP and EFSA [78] also assert that there is no strong evidence for encouraging the delayed introduction of potential food allergens into children’s diets; on the contrary, the early introduction of these foods could be even protective against the development of FAs [79]. For example, the Learning Early About Peanut Allergy (LEAP) study shows that the early introduction (4–11 months) of peanuts into the diets of children who are at high risk of developing FA is effective in reducing the development of peanut allergies, as compared to avoidance [80]. However, these results do not seem to be related to particular changes in the gut microbiota composition but rather to the production of specific IgG4 that plays a protective role against allergy development [80]. Nevertheless, these studies are not conclusive [81] and there are no other convincing studies regarding other foods [80]. It has been proposed that 17 weeks of life is a pivotal time point, with the introduction of complementary feeding before this time appearing to facilitate allergic diseases, whereas solid food introduction after this time seems to promote immune tolerance. This is likely due to the fact that the foods introduced after 17 weeks of age are mainly fruits and vegetables, which are considered to have low allergenic potential and are able to promote non-allergen-specific tolerogenic immunologic mechanisms. Fruits and vegetables are in fact able to induce the maturation of Treg cells via epigenetic mechanisms [82]. Finally, a complementary feeding practice (“a baby-led” approach vs. a “traditional spoon-fed” one) seems to influence the composition of the gut microbiota in children; specifically, baby-led weaning is associated with the lower diversity of the gut microbiota and a reduction in the *Lachnospiraceae* count [72,83], which is related to egg sensitization [84].

## 5. Microbiota Dysbiosis: Mechanisms Associated with FA

Dysbiosis is an imbalance in the gut microbial community; recent data confirm that gut dysbiosis precedes the onset of FA [18,34]. In fact, the allergic response is increased by an early alteration in the gut microbiota composition, associated with a precocious disruption of the gut epithelial barrier [2]. It has been shown that food sensitization is associated with a reduction in gut microbial diversity [35,85], along with an increased abundance of *Enterobacteriaceae* and a decreased abundance of *Bacteroidaceae* and *Ruminococcaceae* [35]. The excessive use of antibiotics is an important risk factor in inducing intestinal dysbiosis, as it alters gut microbial diversity [86]. Gut commensal bacteria play a pivotal role in modulating immune tolerance by reducing circulating basophil populations, promoting epithelial barrier integrity (microbial signals are able to modulate mucous, mucin, and occludin production), and inducing Treg cell differentiation [87]. In an adult mice model, it was demonstrated that gut microbiota dysbiosis is correlated with systemic and local inflammation, leading to intestinal barrier damage. In the same study, a modified microbiota, with a reduction in beneficial bacteria, increased susceptibility to and severity of FA [86]. Recent data indicate that a high-fat diet encourages an increase in allergenic substances, due to an imbalance between “good” and “bad” intestinal bacteria [88]. In particular, a high-fat diet increases total anaerobic microflora and the number of *Bacteroides* spp. and reduces the fecal abundance of *Bifidobacteria* spp. [89,90]. Thus, it is important to create and maintain a symbiosis between humans and their commensal microbiota: if this relationship becomes unbalanced, several pathological processes can occur [2]. The gut microbiota of children affected by FA is characterized by a reduction in *Bacteroides*, *Bifidobacteri*, and *Clostridi* spp., with a consensual abundance in *Anaerobacter* spp. [57,65]. At the same time, Azad et al. [35] demonstrated that there was a low diversity in bacterial species in the gut microbiota of food-sensitized children, along with a relative abundance in *Bacteroides* spp. It has been shown that, in the gut microbiota of infants with FA, there is a reduction in butyrate-producing bacteria [34,54], accompanied by colonization by *Clostridium paraputrificum* and *tertium* [70]. Moreover, there are differences in the gut microbiota composition of children who resolve FAs in the first eight years of life and those who do not: in the first group, *Firmicutes* are most prevalent, while, in the second, there is a higher abundance of *Bacteroidota* [70]. In future, the use of prebiotics and probiotics (i.e., *Lactobacillus rhamnosus*) could help to modulate the gut microbiota composition [54,69], especially in children with a familial history of atopy and in those born by CS [47]. At the same time, innovative approaches, such as the use of synbiotics and fecal microbiota transplantation, could create breakthroughs in this area [91].

In Table 1, we provide a taxonomic classification of the main bacterial phyla, families, and genera involved in allergic diseases.

In Figure 2, we offer a summary of the main protective and unprotective bacterial genera in children’s gut microbiota.

## 6. Conclusions

The first 1000 days of life are a critical period, during which several factors can increase the risk of FA development in genetically predisposed children. Where possible, it is important to try to act on the modifiable risk factors that, in the very early stages of life (pregnancy–infancy–weaning), can identify a “healthy” rather than “unhealthy” intestinal microbiota. Such measures should be undertaken to reduce the incidence of dysbiosis, which is related to the severity of FAs. Breastfeeding is fundamentally important in reducing intestinal inflammation, especially considering HMOs’ prebiotic role. In the future, the use of probiotics and synbiotics could also positively modulate the composition of the gut microbiota, especially in children with a family history of atopy.

## Figures and Tables

**Figure 1 nutrients-15-04014-f001:**
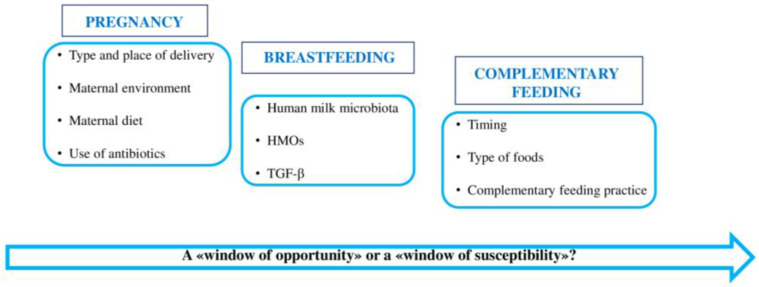
Main factors influencing the composition of children’s gut microbiota in the first 1000 days of life. HMOs: human milk oligosaccharides; TGF-β: transforming growth factor-β.

**Figure 2 nutrients-15-04014-f002:**
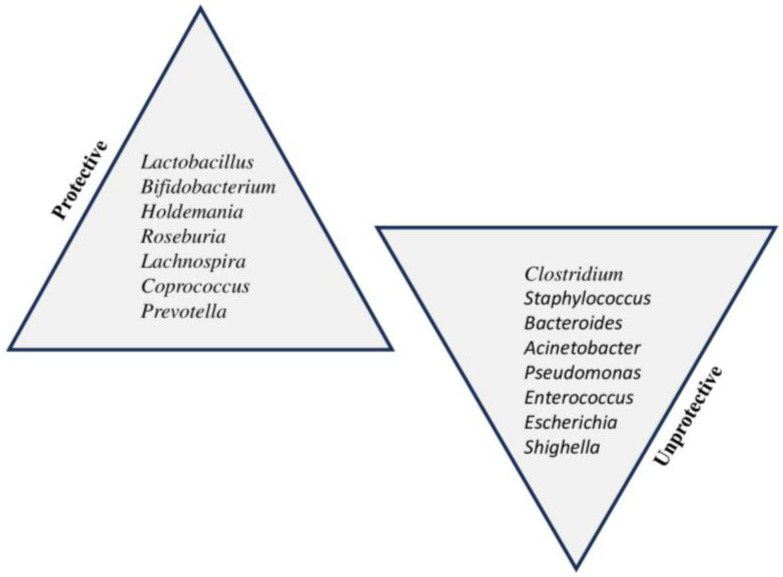
Protective and unprotective bacterial genera in children’s gut microbiota.

**Table 1 nutrients-15-04014-t001:** Taxonomic classification of the main bacterial phyla, families, and genera involved in allergic diseases.

Phylum	Family	Genus
*Firmicutes* [44,54,70]	*Lactobacillaceae*	*Lactobacillus* [8,19,24,41,54,69]
	*Clostridiaceae*	*Clostridium* [8,14,26,35,65,70] *Anaerobacter* [57,65]
	*Staphylococcaceae*	*Staphylococcus* [8,19,61,62]
	*Lachnospiraceae*	*Roseburia* [11] *Lachnospira* [11,54,71,83] *Coprococcus* [11]
	*Streptococcaceae*	*Streptococcus* [61,62]
*Actinomycetota*	*Propionibacteriaceae*	*Propionibacterium* [19,61,62]
*Bacteroidota* [54,70]	*Bacteroidaceae*	*Bacteroides* [26,34,41,57,77]
	*Prevotellaceae*	*Prevotella* [11,19,44,47]
*Actinobacteria* [44]	*Bifidobacteriaceae*	*Bifidobacterium* [2,4,8,14,24,41,54,57,66,67,71,77]
	*Corynebacteriaceae*	*Corynebacterium* [61,62]
*Bacillota*	*Erysipelotrichaceae*	*Holdemania* [11,44]
	*Enterococcaceae* [54]	*Enterococcus* [26,41]
	*Ruminococcaceae* [71]	*-*
*Pseudomonadota*	*Sphyngomonadaceae*	*Sphyngomonas* [61,62]
	*Nitrobacteraceae*	*Bradyrhizobium* [61,62]
*Proteobacteria* [35,44,57]	*Enterobacteriaceae* [35,41,54]	*Escherichia, Shighella [Azad]*
	*Yersiniaceae*	*Serratia* [61,62]
	*Pseudomonadaceae* [44,61,62]	*Pseudomonas* [61,62]
	*Ralstoniaceae*	*Ralstonia* [61,62]
	*Moraxellaceae*	*Acinetobacter* [44]

## Data Availability

Not applicable.

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
