# Peer review of "The First 1000 Days of Life: How Changes in the Microbiota Can Influence Food Allergy Onset in Children"

_nutrients, 2023, doi:10.3390/nu15184014_

Round 1

Reviewer 1 Report

REVIEWER COMMENTS

COMMENTS FOR THE AUTHOR(S)

This is an interesting narrative review of the recent knowledge about the influence of gut microbiota on food allergy onset in children. The authors describe changes in microbiome and their potential relationship with the development of food allergy during the first 1000 days of life.

The manuscript is nicely structured and takes into account: gut microbiota and epigenetics, gut microbiota of pregnant women, in the first months of life, including role of human milk microbiota and human milk oligosaccharides, as well as relationship between gut microbiota and complementary feeding. The authors included interesting table and figure. Although similar articles are available, this is a hot topic worthy of further study.

In my opinion, I see the need for a few changes that could help deliver the ideas/views.

Comments:

1.      Line 60 - please add reference: A dysbiosis of the gut microbiota is, in fact, associated with an increased intestinal permeability, an aberrant inflammatory response and with a different composition of the microbiota.

2.      line 62 - please describe the sentence:  it has been shown a reduction in Lactobacilli and Bifidobacteria that seem to correlate with protection against atopy by favoring Th1 immune response.

3.      The importance of diet in the development of allergic diseases has been repeatedly demonstrated in many articles. The presented manuscript, however, is not strictly about this issue, i.e. diet and allergy, but about the microbiome. So what is missing is an explanation of the potential mechanisms of how diet affects the microbiome. What is more, the subchapter “Microbiota’s dysbiosis: mechanisms associated with FA” does not present the relationship/mechanisms between the microbiome and allergy, how the microbiome may increase or decrease the risk of developing allergies.

4.      Line 215 and line 368 - not only AAP, but also European scientific societies have a similar position, which is worth noting.

5.      Line 259 - please add reference: Atopic dermatitis is associated with lower gut microbiota diversity and relative abundance of Bacteroides spp. by 1 month; an early colonization by Clostridium difficile at 1 month, could be predictive of atopic sensitization at 2 years.

6.      Line 324 - please describe the sentence: Nevertheless, it has been shown that microbial diversity is more beneficial in adults than in children

7.      Line 371 - what is the correlation between LEAP study and microbiome?; the same line 374….

8.      Line 378 - what does it mean: fruit and vegetables, which are not considered allergenic

9.      Line 384 - traditionally - this word seems redundant

10.  Line 391-393 - please add that it concern mice, not humans.

Author Response

This is an interesting narrative review of the recent knowledge about the influence of gut microbiota on food allergy onset in children. The authors describe changes in microbiome and their potential relationship with the development of food allergy during the first 1000 days of life.

The manuscript is nicely structured and takes into account: gut microbiota and epigenetics, gut microbiota of pregnant women, in the first months of life, including role of human milk microbiota and human milk oligosaccharides, as well as relationship between gut microbiota and complementary feeding. The authors included interesting table and figure. Although similar articles are available, this is a hot topic worthy of further study.

In my opinion, I see the need for a few changes that could help deliver the ideas/views.

R: Thanks for your precious suggestions. We attempted to address all the issues raised and we have replied to them point by point. We hope our efforts have improved the quality of the paper and that the revised version can be suitable for publication.

  1. Line 60 - please add reference: A dysbiosis of the gut microbiota is, in fact, associated with an increased intestinal permeability, an aberrant inflammatory response and with a different composition of the microbiota.

R: Thanks for the suggestion. We have done.

  1. line 62 - please describe the sentence:  it has been shown a reduction in Lactobacilli and Bifidobacteria that seem to correlate with protection against atopy by favoring Th1 immune response.

R: We thank the Reviewer for pointing it out. We hope that the sentence sounds good now.

  1. The importance of diet in the development of allergic diseases has been repeatedly demonstrated in many articles. The presented manuscript, however, is not strictly about this issue, i.e. diet and allergy, but about the microbiome. So what is missing is an explanation of the potential mechanisms of how diet affects the microbiome. What is more, the subchapter “Microbiota’s dysbiosis: mechanisms associated with FA” does not present the relationship/mechanisms between the microbiome and allergy, how the microbiome may increase or decrease the risk of developing allergies.

R: Thanks for these comments. Our aim is to describe the correlation between gut microbial composition and FA development. Nevertheless, we have tried to insert, throughout the text (without making a specific paragraph so as not to go off topic) the possible mechanisms by which diet can influence allergy development. Moreover, in the subchapter “Microbiota’s dysbiosis: mechanisms associated with FA”, we have described the principal mechanisms by which microbiota influences the risk of developing allergies: gut microbial diversity, modulation of immune tolerance and influence of epithelial barrier integrity. These mechanisms had been also mentioned and described in the others subchapters. 

  1. Line 215 and line 368 - not only AAP, but also European scientific societies have a similar position, which is worth noting.

R: thanks, we have inserted the new reference.

  1. Line 259 - please add reference: Atopic dermatitis is associated with lower gut microbiota diversity and relative abundance of Bacteroides spp. by 1 month; an early colonization by Clostridium difficile at 1 month, could be predictive of atopic sensitization at 2 years.

R: Thanks for the suggestion. We have done.

  1. Line 324 - please describe the sentence: Nevertheless, it has been shown that microbial diversity is more beneficial in adults than in children

R: We thank the Reviewer for pointing it out. We hope that the sentence sounds good now.

  1. Line 371 - what is the correlation between LEAP study and microbiome?; the same line 374….

R: Thanks for the comment. We have explained in the text that there is not a real correlation between LEAP study and microbiome but probably the results of the trial is due to a specific immunologic production.

  1. Line 378 - what does it mean: fruit and vegetables, which are not considered allergenic

R: We thank the Reviewer for pointing it out. We hope that the sentence sounds good now.

  1. Line 384 - traditionally - this word seems redundant

R: Thanks for the suggestion. We have delated the word above.

  1. Line 391-393 - please add that it concern mice, not humans.

R: Thanks for the comment. We have add it.

Reviewer 2 Report

This is a nice review of the subject of the effect of the microbiome on the development of food allergy. There are many such reviews, which is why I don’t think this paper contributes much to the knowledge of the subject, but it is well presented and reasonably comprehensive.  It would benefit from some minor edits regarding the English language, but in general is pretty good.

Minor edits would be helpful

Author Response

This is a nice review of the subject of the effect of the microbiome on the development of food allergy. There are many such reviews, which is why I don’t think this paper contributes much to the knowledge of the subject, but it is well presented and reasonably comprehensive.  It would benefit from some minor edits regarding the English language, but in general is pretty good.

Comments on the Quality of English Language: Minor edits would be helpful

R: We thank the Reviewer for the suggestion. English language has been checked and partially edited.